# Catalytic disconnection of C–O bonds in epoxy resins and composites

Alexander Ahrens[1✉], Andreas Bonde[1], Hongwei Sun[1], Nina Kølln Wittig[1], Hans Christian D. Hammershøj[1], Gabriel Martins Ferreira Batista[1], Andreas Sommerfeldt[2], Simon Frølich[2], Henrik Birkedal[1] & Troels Skrydstrup[1✉]

Fibre-reinforced epoxy composites are well established in regard to load-bearing applications in the aerospace, automotive and wind power industries, owing to their light weight and high durability. These composites are based on thermoset resins embedding glass or carbon fibres[1]. In lieu of viable recycling strategies, end-of-use composite-based structures such as wind turbine blades are commonly landfilled[1–4]. Because of the negative environmental impact of plastic waste[5,6], the need for circular economies of plastics has become more pressing[7,8]. However, recycling thermoset plastics is no trivial matter[1–4]. Here we report a transition-metal-catalysed protocol for recovery of the polymer building block bisphenol A and intact fibres from epoxy composites. A Ru-catalysed, dehydrogenation/bond, cleavage/reduction cascade disconnects the C(alkyl)–O bonds of the most common linkages of the polymer. We showcase the application of this methodology to relevant unmodified amine-cured epoxy resins as well as commercial composites, including the shell of a wind turbine blade. Our results demonstrate that chemical recycling approaches for thermoset epoxy resins and composites are achievable.

The vast amounts of end-of-use plastics and plastic-containing materials released into nature have resulted in a major environmental crisis[5,6], affecting ecosystems across the globe[9–12]. The necessity for the implementation of a circular economy of plastics and plastic-containing composites has become apparent for reducing the consumption of resources, as well as in limiting the introduction of waste into the environment[5]. In contrast to end-of-use thermoplastics, which can be melted and recast into new forms, the crosslinked polymer chains of thermoset plastics render these materials unsuitable for mechanical recycling. Circumventing processability problems owing to lack of fusibility, chemical recycling deconstructs polymers into their original monomers or related base chemicals that can then re-enter established production chains yielding virgin polymeric materials. Enabling a circular economy in this fashion holds the opportunity of turning accumulating plastic waste into valuable resources[7]. Recently, the catalytic hydrogenation of thermoset polyurethane products for the recovery of anilines and polyols has been reported as a strategy realizing this principle[13,14]. By contrast, epoxy resins lack reactive carbonyl moieties, rendering selective disconnections of their chemical bonds more challenging. Lightweight, highly durable fibre-reinforced epoxy composites, which consist of glass or carbon fibres embedded in the polymer matrix, are high-performance materials crucial for the construction of automobiles, boats, aircraft and wind turbine blades[1]. Wind energy contributed to approximately 6% of the global energy supply as of 2020, with projections forecasting significant growth in the near future[4]. In turn, 43 million metric tons of decommissioned wind turbine blades will have accumulated by 2050 (ref. 15). At the same time, sustainable recycling technologies for such polymeric materials are almost nonexistent. Epoxy resins are not biodegradable and emit toxic gasses on incineration[16], ultimately leading to landfilling as the major pathway for their disposal. As of 2020, only around 1% of end-of-use composites were reused and this by means of shredding the material and using it as a filler substance in construction[1–3]. Because of its inefficiency and unsustainability, landfilling of wind turbine blades has been banned by several European countries, with more being expected to follow[4,17]. Hence, the pressing need for viable recycling strategies for epoxy resins and their composites is mounting[1,4].

Methodologies investigated for recycling of polymer-based composites can be divided into two general approaches, both focused on recovery of fibres only. The first approach relies on destroying the polymer matrix by breaking chemical bonds unselectively, thereby releasing embedded fibres. Reported processes are based on harsh, energy-intense treatments, such as pyrolysis, which is impractical and results in damaged fibres[1–3]. Chemically destructive approaches yield fibres of higher quality[1] but require undesirable reagents such as hydrogen peroxide[18] or concentrated nitric acid[19]. The second, more elegant, approach is to design new epoxy resins containing 'molecular break points', which can be cleaved selectively under certain conditions[20,21]. Although the polymer matrix can be digested into soluble chain fragments, releasing the fibres, the recovered polymer fractions cannot be recast[22–25]. Furthermore, whereas the design of new resins could implement the reuse of fibres for future composite products, the legacy burden of epoxy materials produced up to the present day still remains, as well as those being produced now and in the near future using state-of-the-art resins.

[1]Department of Chemistry and Interdisciplinary Nanoscience Center, Aarhus University, Aarhus, Denmark. [2]Danish Technological Institute, Aarhus, Denmark. ✉e-mail: aahrens@inano.au.dk; ts@chem.au.dk

We envisioned development of a chemical recycling approach for epoxy composites that aims at selective disconnection of the innate linkage motifs shared across epoxy resins, rather than cleaving artificially introduced breaking points or destroying the molecular complexity of the matrix. By targeting bonds formed during basic production steps of the resins, valuable polymer building blocks could be recovered in addition to release of fibres from their polymeric embedment. Thereby, circularity for both epoxy resin and fibres would become achievable because virgin polymers may be produced from recovered base chemicals.

The petrochemical bisphenol A (BPA) represents an important building block for polymers, including epoxy resins[26]. For preparation of the latter, electrophilic epoxide moieties are attached to the BPA backbone via C(alkyl)–O single bonds. Difunctional epoxides can then be cured with multifunctional alkyl amines, yielding randomized three-dimensional (3D) polymer networks knitted together by C–O and C–N σ-bonds in different linkage motifs[27] (Fig. 1a). The potential leakage of BPA into the environment, and the ecological and human health risks associated with such events[28,29], have led to the investigation of potential substitutes[26]. However, with several million metrics tons of BPA-based materials in circulation to date, recovery of these materials and extraction of BPA in a controlled manner are highly desirable and serve to avoid potential leakages from landfilling sites. Biomass-based diphenol or diol motifs could be advantageous replacements for BPA, owing to their renewability[26]. Nonetheless, turning to biorenewable building blocks does not alleviate the requirement for a circular polymer production to minimize its environmental impact[30].

In pursuit of realizing the envisioned approach, we aimed at developing a transition-metal-catalysed method for disconnection of C(alkyl)–O single bonds adjacent to the BPA motif, which form during the reaction of BPA with epichlorohydrin. Carbon–oxygen single bonds have high bond dissociation energies[31,32] and their activation remains challenging. For the valorization of lignin[33], homogenous ruthenium catalysis has been reported to disconnect C(alkyl)–O and C–C single bonds[34–36]. Nonetheless, initial attempts to transfer these methodologies from lignin models to model 1 (Fig. 1a), mimicking one of the most common motifs in epoxy resins, were unrewarding, leading to either low or no conversion (Supplementary Table 1). However, after a comprehensive screening of potential catalysts and reaction conditions (Supplementary Tables 2–6), an efficient protocol was identified. With triphos-Ru-TMM as the precatalyst and three equivalents of isopropanol in toluene at 160 °C, model 1 was cleanly deconstructed providing methylated BPA (Me-BPA) in an isolated yield of 83% (Fig. 1b). No side products were detected from this process. Several other model compounds were tested, adapting the optimized deconstruction conditions. The efficient cleavage of models 2 and 3, yielding Me-BPA at 88% and 80%, respectively, shows that secondary and tertiary amines do not inhibit reactivity, an observation that is crucial when applying the methodology to relevant amine-cured epoxy resins. For substrate model 4 mimicking the less common crosslinked motif 3, no conversion was observed. This linkage model requires the cleavage of three C–O bonds in total, whereby the first C–O bond cleavage is less feasible than for model 1 owing to the more challenging liberation of an alkyl alcohol compared with a phenol. Lastly, model 5, with the central alcohol group capped as a methyl ether, proved unreactive to the deconstruction conditions.

Beyond linkage motifs, we considered model compounds containing bisphenol or diol scaffolds other than BPA (Fig. 1b). Bisphenol S (BPS) is a commercially relevant compound with a sulfonyl linker in its backbone, resulting in different electronic properties. The BPS-based model 6 reacted cleanly under optimized conditions, allowing for a 74% isolated yield of methylated BPS. Next, three other model compounds containing potential biorenewable substitutes for BPA[26] were investigated. We tested our disconnection approach on a representative selection of such candidates. Model 7 is based on a bisphenol obtained from

the terpenoid, carvacrol[37,38], that can be synthesized from limonene[39]. Under catalytic conditions, efficient and selective C–O bond cleavage was observed. Vanillin-derived compounds can be sourced from lignin, making them attractive candidates for replacement of petrochemical sources. An epoxy model based on p,p'-bisguaiacol F (model 8)[40] was subjected to catalytic deconstruction conditions and, after 16 h, methylated p,p'-bisguaiacol F was isolated at 57% yield. Although it can be speculated that the more sterically encumbered C–O bond is less accessible to the catalyst, the reaction is highly selective because only unreacted starting material was recovered apart from the cleavage product. Lastly, a bis(hydroxymethyl)furan-based model compound (model 9) was tested[41]. Furan-related building blocks are of interest because of their availability from cellulose[42]. No conversion was observed, showing that Ru-catalysed cleavage is selective for phenol-based linkages. With this scope of model substrates at hand, studies on catalytic activity in model 1 were conducted. Even at a reduced catalyst loading of 0.38 mol%, quantitative conversion to Me-BPA was observed after 16 h. This corresponds to a turnover number (TON) of 533.

As supported by the inertness of model 5 to the catalytic conditions, and in line with mechanisms proposed for cleavage on lignin β-O-4 linkage models[34,36], we postulate that C–O aryl bond breakage is preceded by dehydrogenation of the alcohol functionality in model 1, forming ketone 1 (Fig. 2a). The bond dissociation energies (BDEs) for the C–O ether bonds of model 1 and the corresponding ketone (ketone I) were calculated. In alignment with theoretical studies on lignin linkages[31], the BDE of the latter is 10.7 kcal mol$^{-1}$ lower (Fig. 2a,b), supporting the premise that dehydrogenation is necessary for initiation of C–O activation.

From ketone I, we propose that C–O single bonds are susceptible to cleavage via an oxidative addition step involving a low-valent ruthenium complex eventually generating Ru-II (Fig. 2b). Through a dehydrogenation step with isopropanol, intermediate Ru-II is reduced with subsequent formation of species Ru-III and the liberation of a phenol. The low-valent ruthenium complex Ru-III undergoes a second oxidative addition step, and the reduction cascade ultimately leads to cleavage of the model substrate into acetone and phenol components. The kinetic profile of bond disconnection on model 1 was investigated (Fig. 2d) showing an induction period of approximately 2 h, after which only a minute amount of Me-BPA could be detected. However, after 4 h, a 50% yield of Me-BPA was observed with quantitative conversion reached after 12 h. An induction period hints at catalyst activation preceding the catalytic cycle. At a reaction time of 4 h, traces of the monocleavage intermediate ketone III were detected. In that, otherwise, only the starting material and product were observed indicates that the consumption of intermediates is significantly faster than that of model 1. Density functional theory (DFT) calculations showed that the C–O BDE of ketone I is 3.3 kcal mol$^{-1}$ lower than that of ketone III (Fig. 2b), implying that the more rapid consumption of the latter is not controlled by C–O bond strength. Possibly the second C–O bond activation is more rapid than dissociation of the intermediate from the catalyst. Ketone III was subjected to catalytic conditions in a one-to-one mixture with model 1. Both compounds were consumed yielding Me-BPA, which supports ketone III being an intermediate. To gather further support for the proposed C–O bond-cleaving mechanism, efforts were undertaken to detect acetone as the corresponding disconnection product formed from the central linkage motif of model 1 (Fig. 2c). Because the dehydrogenation of isopropanol yields acetone, the hydrogen source was exchanged for 1-phenylethanol. After 16 h of reaction time, both a quantitative formation of acetophenone from 1-phenylethanol and 74% yield of acetone from the linkage motif were observed.

The deconstruction of model 1 using 3 mol% triphos-Ru-TMM in toluene at 160 °C in the absence of isopropanol yielded only 4% of Me-BPA after 16 h (Supplementary Table 1). If isopropanol serves only as an additional hydrogen source, approximately 50% of the C–O bonds should have been cleaved. To investigate the role of isopropanol,

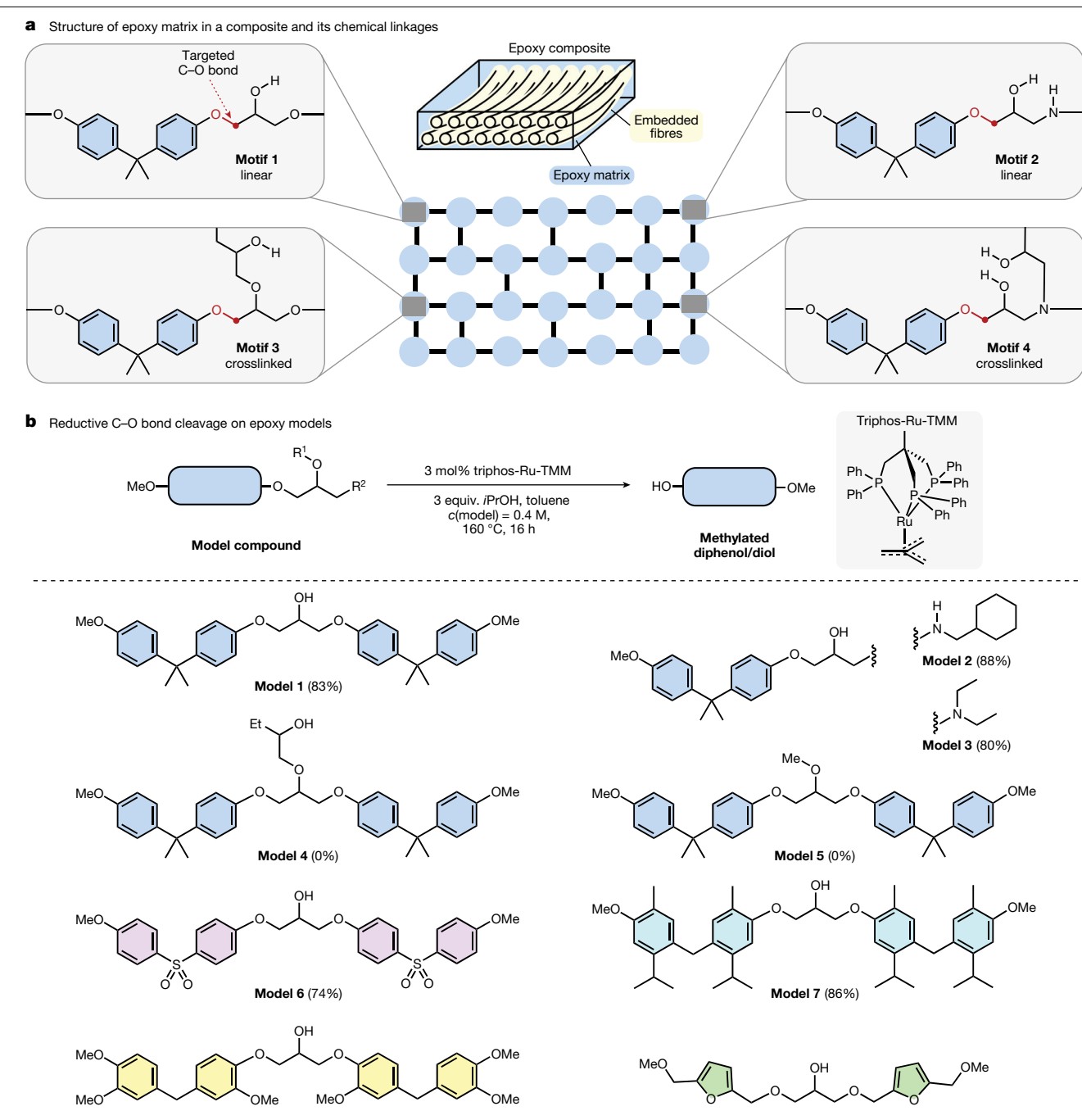

**a** Structure of epoxy matrix in a composite and its chemical linkages

Targeted C–O bond

**Motif 1** linear

**Motif 2** linear

**Motif 3** crosslinked

**Motif 4** crosslinked

Epoxy composite

Embedded fibres

Epoxy matrix

**b** Reductive C–O bond cleavage on epoxy models

Triphos-Ru-TMM

3 mol% triphos-Ru-TMM

3 equiv. *i*PrOH, toluene
*c*(model) = 0.4 M,
160 °C, 16 h

**Model compound** → **Methylated diphenol/diol**

**Model 1** (83%)

**Model 2** (88%)

**Model 3** (80%)

**Model 4** (0%)

**Model 5** (0%)

**Model 6** (74%)

**Model 7** (86%)

**Model 8** (57%)

**Model 9** (0%)

**Fig. 1 | Targeted C–O bonds in thermoset epoxy resins and catalytic deconstruction of related model compounds. a**, Schematic illustration of a crosslinked epoxy resin matrix and molecular structures of linkage motifs. Blue circles represent linkage sections while black lines represent linear polymer sections. The C–O bonds adjacent to BPA (red) are targeted to deconstruct the polymer matrix. **b**, Optimized reaction conditions applied to different model substrates, considering linkage motifs and building blocks. Yields are given (in parentheses) for products isolated via column chromatography.

operando monitoring experiments were conducted (Fig. 2d) using [1]H and [31]P nuclear magnetic resonance (NMR) spectroscopy. The experiment, in the absence of isopropanol, was run in a J. Young NMR tube. After 16 h, no conversion of the starting material was observed. Likewise, triphos-Ru-TMM had not been consumed and no other ruthenium species could be detected. The reaction was repeated with 3 equiv. of isopropanol present. Here, close to quantitative conversion of the starting material to Me-BPA was observed after 16 h. In addition, the starting ruthenium complex was consumed and the formation of new signals corresponding to ruthenium species was detected in both the [31]P NMR spectra and the hydride region of the [1]H NMR spectra. The addition of

another equivalent of **model 1** to the reaction mixture containing the new ruthenium species and running it again led to a clean deconstruction of the fresh substrate into methylated BPA.

A phosphine singlet peak at 47.3 ppm could not be identified. A hydride-bridged binuclear ruthenium(I) complex[43] was detected and tested as a potential precatalyst on **model 1**. With no conversion being observed, the formation of this species can be considered a deactivation pathway. Lastly, a triplet peak and doublet phosphine peak could be linked to a hydride signal using an [1]H, [31]P heteronuclear single quantum coherence NMR experiment. Single crystals suitable for X-ray crystallography were obtained, allowing the identification of this species as

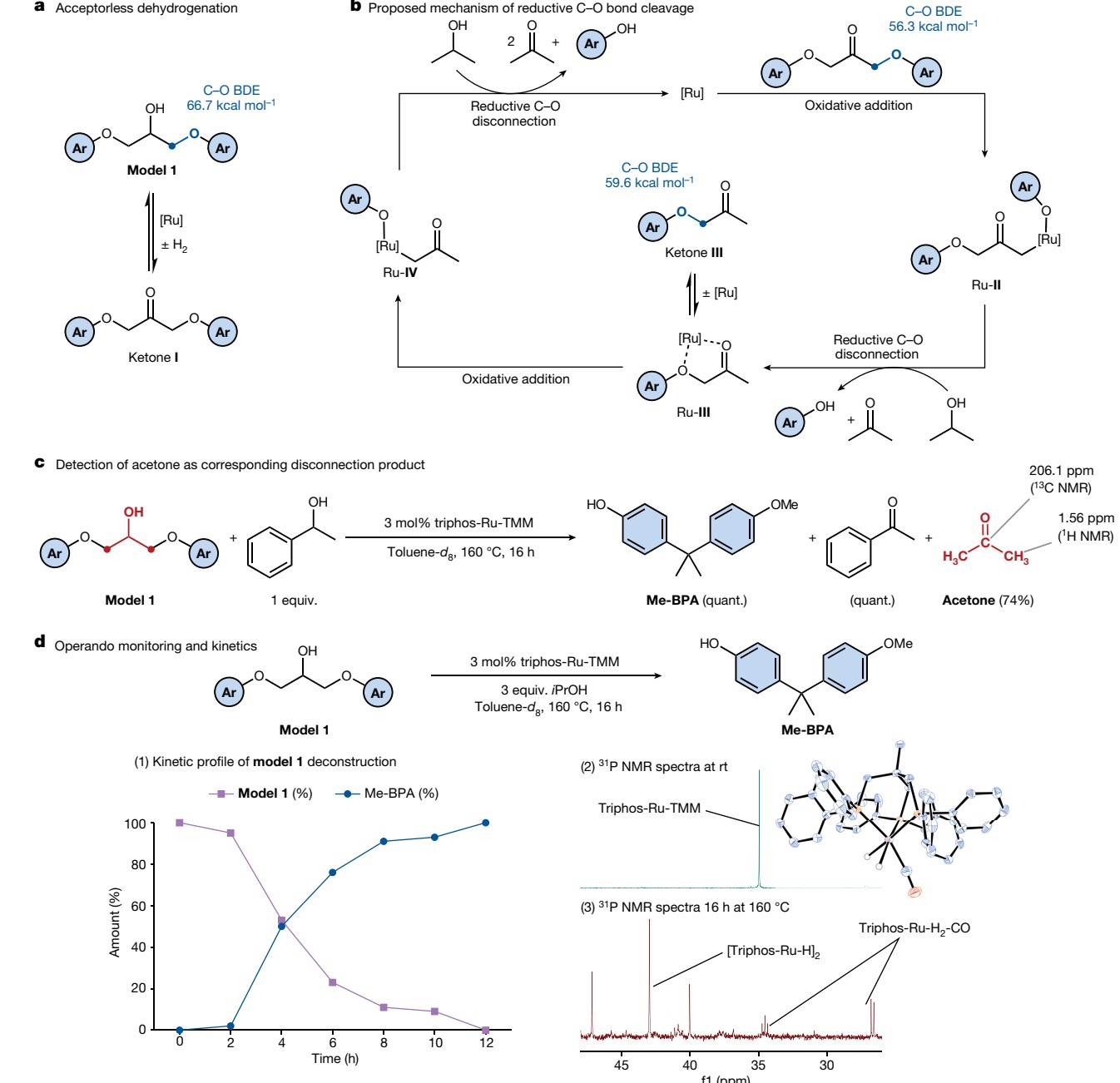

**a** Acceptorless dehydrogenation

C–O BDE
66.7 kcal mol⁻¹

**Model 1**

[Ru]
± H₂

Ketone **I**

**b** Proposed mechanism of reductive C–O bond cleavage

Reductive C–O
disconnection

[Ru]

C–O BDE
56.3 kcal mol⁻¹

Oxidative addition

Ru-**II**

Reductive C–O
disconnection

Ru-**IV**

C–O BDE
59.6 kcal mol⁻¹

Ketone **III**

± [Ru]

Ru-**III**

Oxidative addition

**c** Detection of acetone as corresponding disconnection product

**Model 1**        1 equiv.

3 mol% triphos-Ru-TMM

Toluene-$d_8$, 160 °C, 16 h

**Me-BPA** (quant.)        (quant.)        **Acetone** (74%)

206.1 ppm
(¹³C NMR)

1.56 ppm
(¹H NMR)

**d** Operando monitoring and kinetics

**Model 1**

3 mol% triphos-Ru-TMM

3 equiv. $i$PrOH
Toluene-$d_8$, 160 °C, 16 h

**Me-BPA**

(1) Kinetic profile of **model 1** deconstruction

— **Model 1** (%)    — Me-BPA (%)

(2) ³¹P NMR spectra at rt

Triphos-Ru-TMM

(3) ³¹P NMR spectra 16 h at 160 °C

[Triphos-Ru-H]₂        Triphos-Ru-H₂-CO

**Fig. 2 | Mechanistic considerations regarding Ru-catalysed C–O bond disconnection. a**, Ru-catalysed acceptorless dehydrogenation. **b**, Proposed catalytic cycle for disconnection of C–O bonds. BDEs were calculated with DFT at the (U)M06-2X/6-311++G(d,p) level of theory. **c**, Detection of acetone as the disconnection product. Yields were determined by ¹H NMR spectroscopy using 1,3,5-trimethoxybenzene as the internal standard. **d**, Kinetic profile of deconstruction of **model 1**. Operando monitoring experiment for C–O bond disconnection on **model 1**. Molecular structure of triphos-Ru-H₂-CO in the crystal (CCDC 2219777).

triphos-Ru-H₂-CO. We propose that the CO originates from acetone, which presents a rare case of decarbonylation of a secondary alcohol[44]. The carbonyl complex was tested as precatalyst, but proved inactive.

With the identification of a suitable catalytic system for the deconstruction of epoxy models, we moved on to a polymeric resin (Fig. 3). Airstone 760E/766H is a thermoset epoxy system adapted to the construction of wind turbine blades, consisting of four monomers and containing approximately 43 wt% of BPA after curing. Initially a 'dogbone' of the cured resin was ground into powder, suspended in toluene-containing isopropanol and stirred at 160 °C in the absence of catalyst (entry 1). After 24 h, no compounds liberated from the resin could be detected. By contrast, when 6 wt% of catalyst was added,

a 56% yield of BPA was isolated (entry 2). Both gas chromatography–mass spectrometry and ¹H NMR spectroscopic analysis of the BPA sample did not show any detectable impurities, supporting the potential for its reuse. Furthermore, a highly polar rest fraction was recovered, the analysis of which by matrix-assisted laser desorption/ionization time-of-light mass spectrometry, ¹H NMR and infrared spectroscopy showed a complex mixture of oligomers containing alkyl ethers, amines and small amounts of aromatics. In total, 81 wt% of the cured resin was recovered as soluble organic material. The influence of particle size on the efficiency of the deconstruction suggests that catalysis is limited to the surface area of the resin particles suspended in solution (Supplementary Table 7). An attempt to decrease catalyst loading

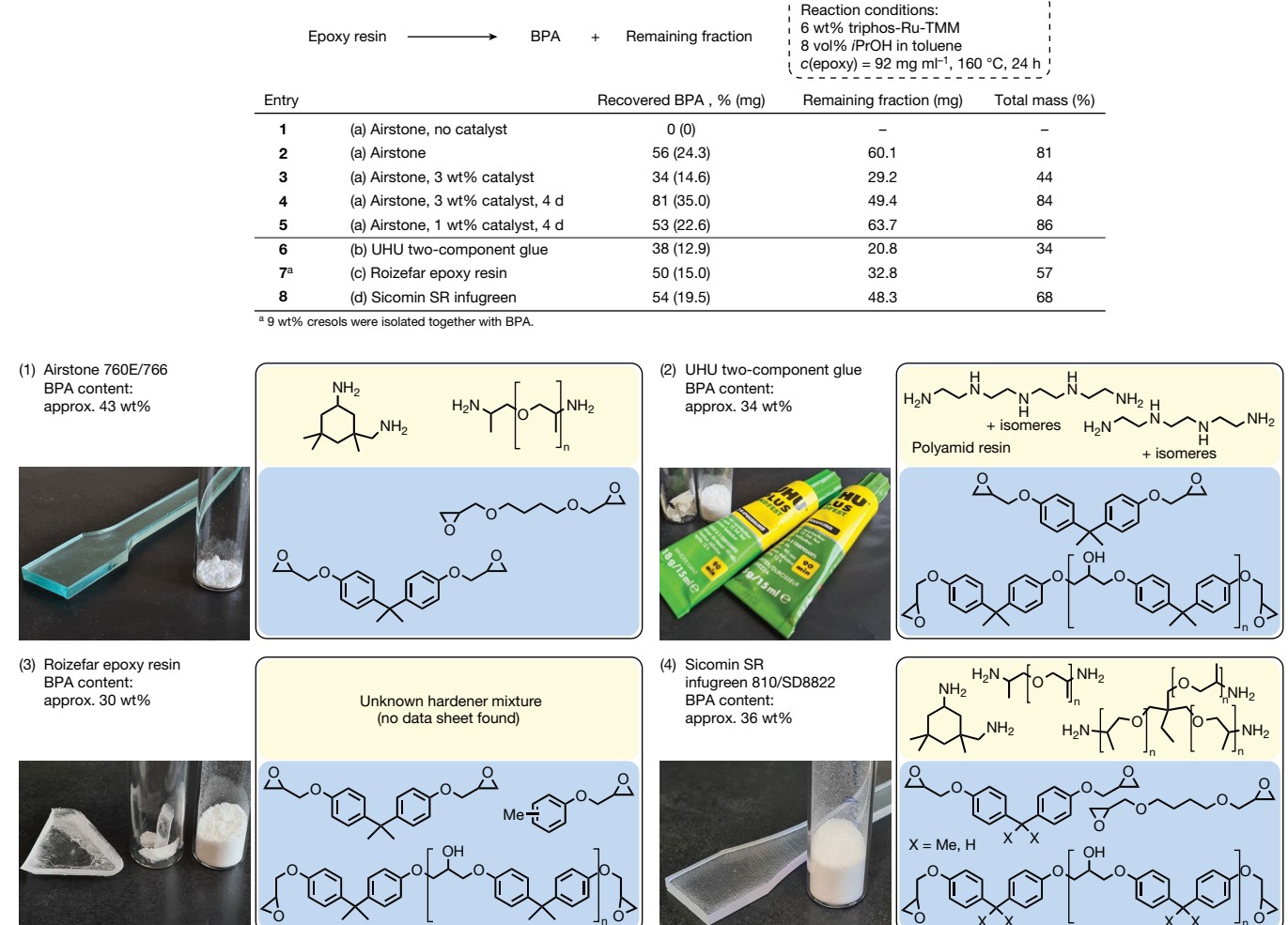

| Entry | | Recovered BPA , % (mg) | Remaining fraction (mg) | Total mass (%) |
|---|---|---|---|---|
| **1** | (a) Airstone, no catalyst | 0 (0) | – | – |
| **2** | (a) Airstone | 56 (24.3) | 60.1 | 81 |
| **3** | (a) Airstone, 3 wt% catalyst | 34 (14.6) | 29.2 | 44 |
| **4** | (a) Airstone, 3 wt% catalyst, 4 d | 81 (35.0) | 49.4 | 84 |
| **5** | (a) Airstone, 1 wt% catalyst, 4 d | 53 (22.6) | 63.7 | 86 |
| **6** | (b) UHU two-component glue | 38 (12.9) | 20.8 | 34 |
| **7**[a] | (c) Roizefar epoxy resin | 50 (15.0) | 32.8 | 57 |
| **8** | (d) Sicomin SR infugreen | 54 (19.5) | 48.3 | 68 |

Reaction conditions:
6 wt% triphos-Ru-TMM
8 vol% $i$PrOH in toluene
$c$(epoxy) = 92 mg ml$^{-1}$, 160 °C, 24 h

Epoxy resin $\longrightarrow$ BPA + Remaining fraction

[a] 9 wt% cresols were isolated together with BPA.

**Fig. 3 | Catalytic deconstruction of epoxy resins.** Scope of epoxy resins deconstructed using catalytic conditions. Experiments were set up under an argon atmosphere. Yields were determined after isolation of products via column chromatography.

by 50% to 3 wt% (entry 3) reduced the amount of recovered BPA to a 34% yield. However, extending the reaction time to 4 days (entry 4) increased the amount of recovered BPA to 81%, showing that the catalyst was not deactivated after 24 h. Another 4-day experiment using 1 wt% of catalyst provided a 42% yield of BPA corresponding to a TON of 105 (entry 5).

Three additional commercial epoxy resins were tested. A two-component adhesive (entry 6), containing a complex mixture of multifunctional hardeners resulting in a higher degree of crosslinking, proved to be more challenging to deconstruct. Nonetheless, a 38% yield of BPA was recovered after 24 h. A clear-cast epoxy system marketed for handicrafts (entry 7) was also disassembled yielding 50% of BPA. Furthermore, this resin was found to contain a cresol-based epoxy component because two isomers of cresol were recovered at a yield of 9 wt% together with BPA. Next, a partially biomass-based infusion system for maritime engineering was tested (entry 8). This system comprises a complex mixture of multifunctional hardeners and contains small amounts of bisphenol F (BPF) epoxies. Although 54% of the BPA was recovered from this sample, the separated BPF remained elusive after deconstruction possibly because of the modest amounts of BPF epoxies applied in this specific mixture. Last, Lightstone 3100E/3102H, an anhydride-curing system developed for pultrusion applications, was subjected to catalysis. However, for this sample no BPA could be isolated. Anhydride-curing results in the formation of linkage motifs different from those derived from amine curing, because the alcohol

moieties of the epoxy fraction are acylated[27]. This functionalization blocks dehydrogenation and subsequently C−O bond cleavage analogously to **model 5**.

With a general method available for molecular disassembly of amine-cured epoxy resins, we turned to investigate the suitability of this protocol for the deconstruction of fibre-reinforced epoxy composites, which, apart from the polymer matrices, contain a high weight percentage of fibres. For recovery of the latter, powderizing epoxy composites are counterproductive. We recognized that composite materials may be more accessible to the solvent because of the interface area between fibres and polymer. Our work commenced with a carbon fibre-based composite ((1) in Fig. 4a), procured from landfill. With no previous treatment apart from cutting to size, a cube of material was submerged in the solvent mixture, 6 wt% of catalyst was added and the mixture stirred at 160 °C. After 3 days the composite had visibly separated into loose fibres. The reaction mixture was decanted; after washing, 57 wt% of carbon fibres was recovered and, from the solution, 13 wt% of BPA was isolated. Although we could not estimate the total amount of BPA in the material, the complete solubilization of the polymer matrix and mass balance point towards an efficient recovery. Next, a commercial product sample of a glass fibre-based laminate ((2) in Fig. 4a)) was treated in the same manner. Here too visible separation into loose fibres was observed over the course of 3 days. This composite yielded 53 wt% of glass fibres and 15 wt% of BPA, demonstrating that the method does not differentiate between glass and carbon

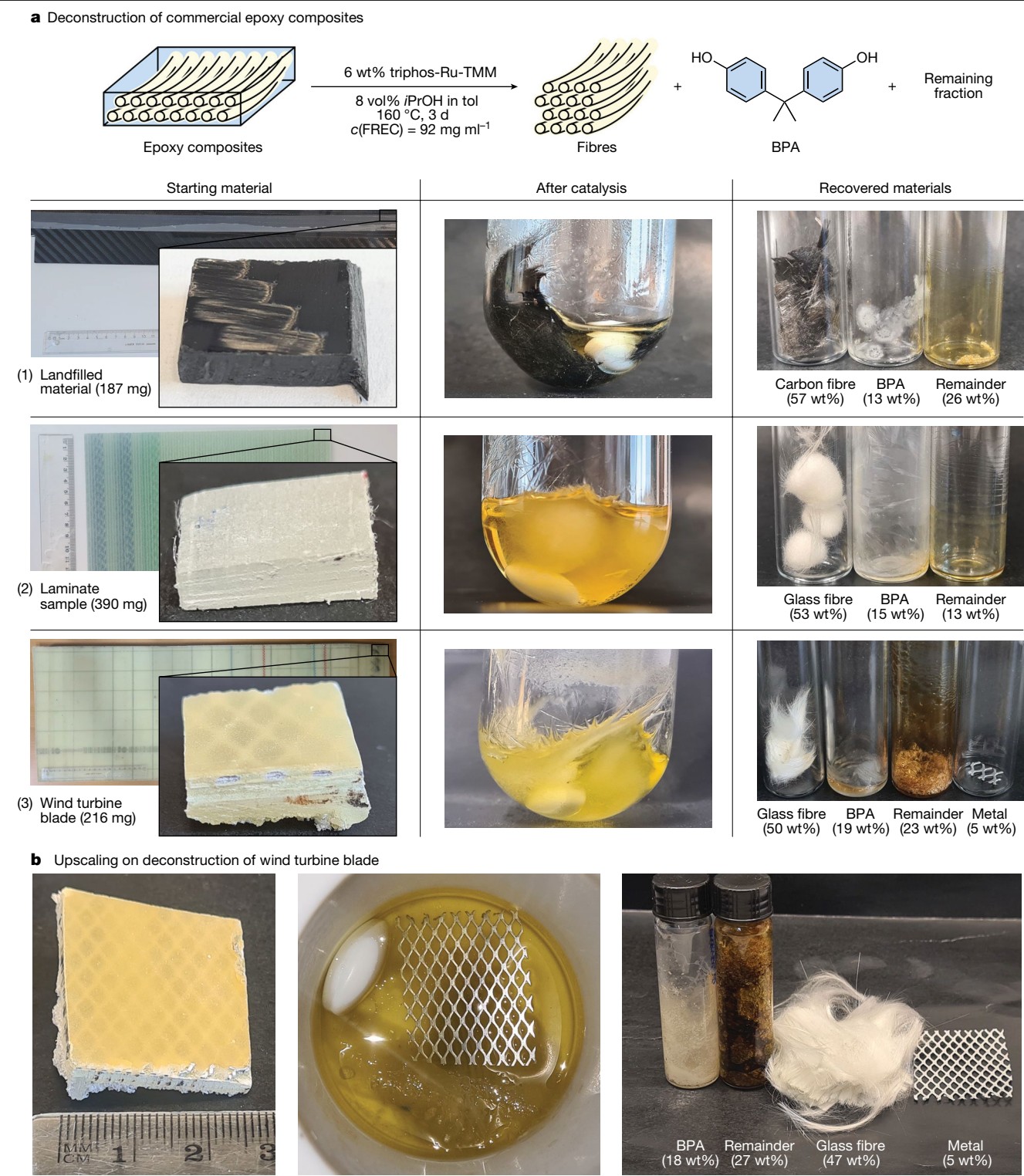

**Fig. 4 | Recovery of BPA and fibres from commercial epoxy composites using Ru catalysis. a**, Scope of the composite samples subjected to catalysis. Composite pieces 1, 2 and 3 were 1.0–1.5 cm in both length and width. **b**, Upscaling of deconstruction conditions on wind turbine blade.

fibre-based composites. Last, with these promising results at hand, a piece of the outer shell of a state-of-the-art decommissioned wind turbine blade ((3) in Fig. 4a) was tested. This commercial composite sample was cleanly disassembled, yielding 50 wt% of glass fibres and 19 wt% of BPA. Additionally a piece of metal grid, incorporated in the blade as part of the lightning protection system, was separated from the structure.

Finally, the possibility of scaling up the catalytic protocol to larger pieces of composite material was examined (Fig. 4b). A 5.13-g plate of decommissioned wind turbine blade was placed in a 300-ml autoclave and subjected to catalytic conditions. After 6 days of reaction time the matrix had been fully disassembled, allowing the isolation of 4.0 mmol of pure BPA and the recovery of 2.4 g of pristine glass fibres. Furthermore, a sheet of metal grid was also recovered.

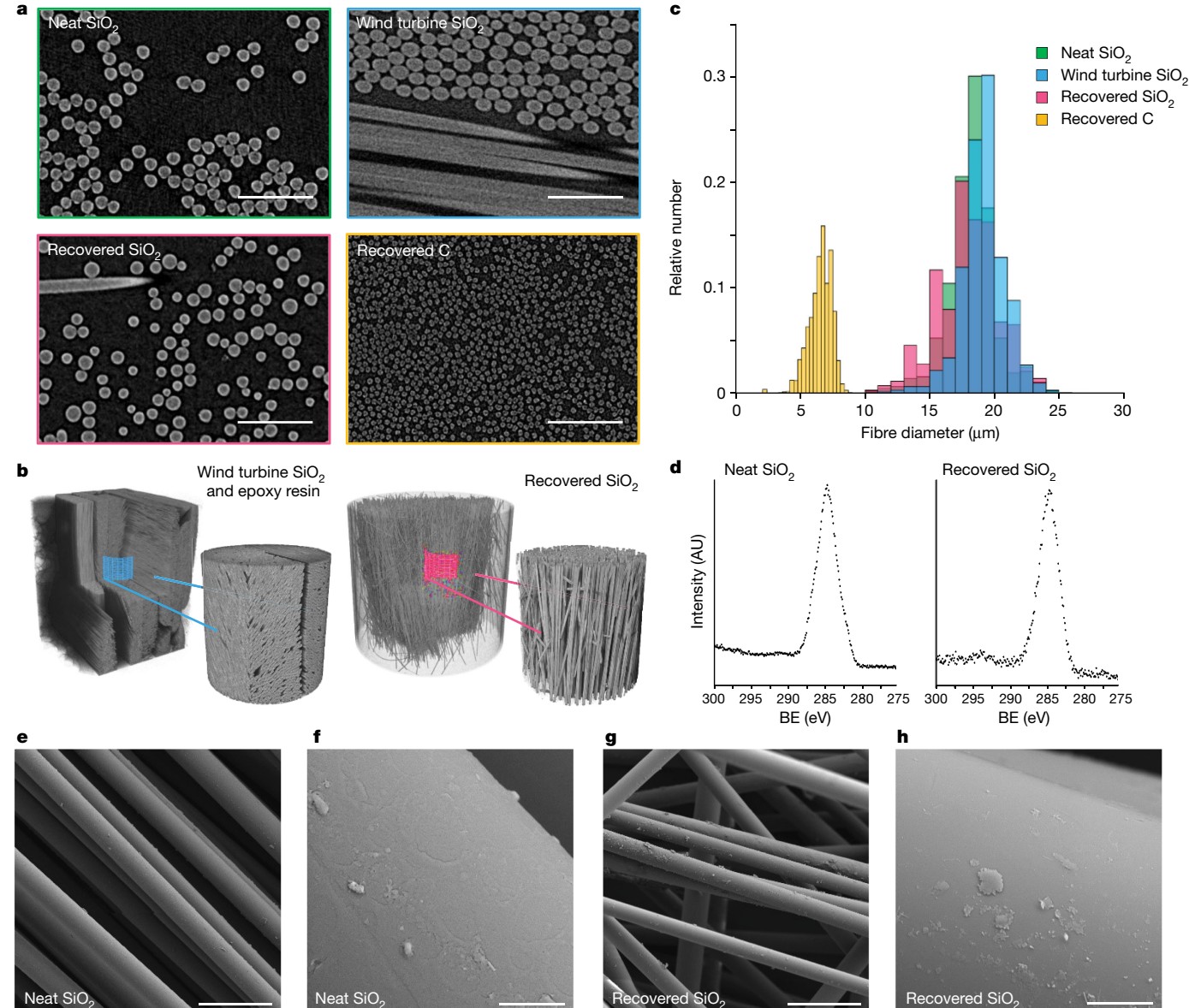

**Fig. 5 | Characterization of commercial fibre-reinforced composites and glass fibres. a**, X-ray μ-CT with virtual slices through reconstructed image stacks showing fibre cross-sections. Scale bars, 100 μm. **b**, 3D renderings of reconstructed image stacks showing fibre organization; grey levels corresponding to air have been rendered transparent; for scale refer to the two-dimensional slices in **a**. **c**, Histograms of fibre diameter obtained by analysis of X-ray μ-CT data. **d**, XPS C 1s high-resolution spectra of neat and recovered fibres. **e**–**h**, SEM images of neat (**e**,**f**) and recovered fibres (**g**,**h**). Scale bars, 50 μm (**e**,**g**); 2 μm (**f**,**h**). AU, arbitrary units; BE, binding energy.

To evaluate the quality of the recovered fibres in comparison with neat fibres, X-ray micro-computed tomography (μ-CT), X-ray photoelectron spectroscopy (XPS) and scanning electron microscopy (SEM) were used (Fig. 5). With μ-CT, grey-level variations in the images reflect varying material density. Glass fibres appeared lighter than the epoxy resin, which, in turn, appeared slightly lighter than air (Fig. 5a). Figure 5b shows 3D renderings of fibres where air has been made transparent. Epoxy resin was observed only in the nontreated piece of the wind turbine blade, and the images visually corroborate that the high quality of the recovered fibres was preserved. Diameters of the glass fibres were quantified and found to be similar to those of neat glass fibres (18 ± 2 μm), and for glass fibres both embedded within (19 ± 2 μm) and recovered from (18 ± 2 μm) a piece of decommissioned wind turbine blade. The carbon fibres salvaged from landfilled material (6.5 ± 0.9 μm) were substantially smaller (Fig. 5c).

X-ray photoelectron spectroscopy was used to test whether the epoxy resin had been completely removed from the fibres in the deconstruction process (Supplementary Table 9). The atomic concentrations of Si, Ca and Al relative to C were higher in the recovered than in the neat fibres. The higher relative C content of the latter originates from the priming layer used to coat glass fibres, which was partially removed during catalysis. No residual polymer was detected, and this was further corroborated by the high-resolution C 1s spectra of both neat and recovered fibres (Fig. 5d), for which the π-π* type shake-up peaks typically detected for C in aromatic compounds (around 291–292 eV) were absent. SEM images of fibres show the imprint of this coating on neat fibres (Fig. 5e,f and Supplementary Fig. 16a–d,i), whereas the surface of the recovered fibres is smooth (Fig. 4g,h and Supplementary Fig. 16e–h,j). Finally, preliminary tensile strength studies on fibres recovered from the wind turbine blade, with neat fibres as the reference point, showed comparable mechanical strength (Supplementary Table 10).

For those components recovered from end-of-use composites, perspectives for circularity can be considered. The high purity of the BPA recovered allows, in principle, its reuse in established production chains for epoxy resins, polycarbonates or polyesters, replacing virgin BPA produced from naphtha feedstock. The remaining fraction, consisting of various oligomers, unfortunately cannot be used as chemical building blocks. Nonetheless, valorization strategies beyond energy recovery can be envisioned. For example, pyrolysis has been demonstrated to process mixed plastic wastes (including nitrogen-containing polymers) into, for example, naphtha equivalents or syngas[45,46]. As such, this remaining fraction may find uses as a carbon feedstock source for the chemical industry. Last, in regard to the glass and carbon fibres recovered at high quality, several reuse approaches have been reported. These include the use of recovered fibres to construct new composite materials after a chemical sizing or resizing process[47,48]. With these considerations in mind, our catalytic process can be considered as a proof-of-concept demonstration that a circular economy may well be achievable for these valuable and relevant materials.

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

# Article

## Methods

Descriptions of the methods used are provided in the Supplementary Information.

## Data availability

Crystallographic data are available free of charge from the Cambridge Crystallographic Data Centre under no. CCDC 2219777. The authors declare that all other data supporting the findings of this study are available within the paper and its Supplementary Information files.

**Acknowledgements** We thank the entire CETEC consortium—in particular M. E. Birkbak from Vestas Wind System A/S, S. C. A. Lauth, M. Schrötz and L. O. Meyer from the Olin Corporation, as well as E. Damgaard-Møller and A. S. Donslund from the Danish Technological Institute— for valuable discussions, epoxy resin/composite material samples and assistance regarding analytics. We thank M. Ceccato from Aarhus University for measurement of XPS spectra and S. S. Pedersen from Aarhus University for procuring a carbon fibre composite sample. We also thank CSCAA for the computing hours provided for the DFT study. We are deeply grateful for financial support by Innovation Fund Denmark (grant no. 0224-00072B), Carlsberg Foundation (grant no. CF18-1101), Danish National Research Foundation (grant nos. DNRF118 and DNRF-93), Novo Nordisk Foundation research infrastructure grant for AXIA (no. NNF19OC0055801) and Aarhus University. Support from the European Union's Horizon 2020 research and innovation programme under grant agreement no. 862179 and Marie Sklodowska-Curie grant agreement no. 859910 is also gratefully acknowledged. This publication reflects the views only of the authors, and the Commission cannot be held responsible for any use that may be made of the information contained therein.

**Author contributions** Conceptualization, writing and revising of the original draft were carried out by A.A. and T.S. Experimental design was carried out by A.A. Experimental investigation was carried out by A.A., A.B. and H.S. A.A. and T.S. supervised and directed research. Analytics and evaluation of recovered fibres were conducted by N.K.W., H.B., A.S. and S.F. X-ray crystallographic investigations were conducted by H.C.D.H. DFT studies were conducted by G.M.F.B. Preparation of clear-cast epoxy samples and mass analysis on remaining fraction were carried out by A.S. Funding acquisition was carried out by T.S. and S.F. All authors reviewed the final manuscript.

**Competing interests** A.A., T.S., A.S. and S.F. are inventors on provisional patent application no. EP22156129, submitted by Aarhus University, which covers the transition-metal-catalysed disassembly of epoxy-based, fibre-reinforced, polymer composites. T.S. is co-owner of SyTracks A/S, which commercializes CO tubes.

**Additional information**
**Correspondence and requests for materials** should be addressed to Alexander Ahrens or Troels Skrydstrup.
