## [Peer Review File · Nature]

Manuscript Title: Catalytic Disconnection of C–O Bonds in Epoxy Resins and Composites

Editorial Notes:

Reviewers #2 and #3 did not respond to requests to further review the work. Reviewer #4 was added to supplement the necessary expertise to assess the revised manuscript, and Reviewer #1 was also asked for their feedback on the revisions as a whole.

Reviewer Comments & Author Rebuttals

Reviewer Reports on the Initial Version:

Referee #1 (Remarks to the Author):

Ahrens, Skrydstrup and coworkers disclosed a viable strategy of catalytic disconnection of C–O bonds in epoxy resins and composites. Epoxy resins, as one of the most important thermosets, have been widely used in our life, but they are difficult to be recycled due to their permanently cross-linked networks. With the increasing consumption of composites, especially from wind energy, light-weight automobile and aviation, the consumption of epoxy resins is mounting. Now, the epoxy resins and their composites are simply utilized as fillers by mechanically grinding, incinerated for energy recovery, or landfilled. And they are also recycled via pyrolysis and solvolysis to obtained gas (or liquid) fuels and oligomers (or monomers), respectively. But for epoxy resins, especially amine-cured ones, it's still a challenge to recover the original monomers or raw materials though oligomers with specific structures could be achieved. In this work, authors found a catalytic system which could selectively disconnect the C-O bonds. The selective disconnection reactions and catalytic mechanisms were systematically investigated by model reactions. Besides, the catalytic systems could be used to degrade the cured epoxy resins and their composites, and excitingly, the raw material bisphenol A can be achieved. Because bisphenol A epoxy resins account for around 90% of the total output of epoxy resins, the recovery of bisphenol A is extremely beneficial to promote the sustainable development of epoxy resins and composites. After careful consideration, I think this work should be published in Nature. I believe if this work is published in Nature, more researchers will make efforts in recycling epoxy resins and composites. Before publication, some improvements are necessary. The comments and suggestions are as follows.

1. The literatures on selective cleavage of epoxy resins and their composites should be summarized and discussed in the introduction section of the manuscript. Otherwise, the science and technology advances of this work will be not clear for readers.
2. The chemical structure of benzophenone is much different from ketone 1. Whether more suitable model compound could be used to investigate the degradation mechanism? From the degradation mechanism proposed by authors, ketone should be produced, so it's better to directly capture the ketone during the degradation process.
3. For the real amine-cured epoxy networks, there should be a large amount of tertiary amines and a small amount of secondary amines. Thus, beside the model 2 with secondary amines, model with tertiary amines should be investigated. And for the anhydride-cured epoxy resins, beside the C-O

bonds, there are a lot of ester bonds, do the ester bonds impact the disconnection efficiency of C-O bonds?

4. The efficiency for the catalytic disconnection of C-O bonds is relatively low. Can it be improved ?

5. For evaluation of the advance in disconnection of composites, beside the micromorphology and appearance of the reclaimed fibers, the chemical structures and the mechanical properties of the reclaimed fibers should be examined and compared with the virgin fibers. And it's better to compare with those from other recycle methods using sever conditions.

6. The peaks in each NMR spectrum should be clearly matched with the chemical structures of the compounds.

7. The stability of the catalysts during the reactions and their reuse should be investigated.

8. The disconnection yield after certain times were provided, but the kinetics of the disconnections were not systematically investigated, as a result, readers cannot see the balance point of the degradation reactions.

9. Authors used iPrOH and toluene as the solvent, I think different solvents might have very different degradation rate even degradation mechanism. Therefore, the effects of different solvents on the disconnection should be investigated.

Referee #2 (Remarks to the Author):

In the present manuscript by Ahrens and Skrydstrup, a catalytic system for the depolymerisation of selected epoxy resins is described. The recycling of polymers is a field of current scientific interest and practical systems for the sustainable depolymerisation of complex materials remain rather elusive. The present approach tries to use molecular ruthenium catalyst systems for the effective catalytic disconnection of C–O Bonds in polymeric materials based on epoxy resins.

The authors analysed the chemical structure of the epoxy resins and aimed at developing a transition metal catalysed method for disconnecting the C(alkyl)–O single bonds. The model compounds used by the authors are very similar to lignin model compounds, for which the cleavage of the C-O bonds has already been demonstrated. Especially in references 37 and 38 (incl. SI) the different bond cleavages and products are clearly broken down. The reference to these publications becomes even clearer when one recognizes that the catalyst Ru(triphos)tmm and its derivatives were also used in this preliminary work. Surprisingly, the authors avoid making a clear reference to this previous work and justify their own contribution with the low activity of the already published catalytic systems with the catalyst Ru(triphos)tmm. The main modification in the present manuscript is the addition of iPrOH, but the contribution of the secondary alcohol is not mechanistically proven by the present experiments, only simple dehydrogenation experiments are shown. In addition, the proposed mechanism (FIG. 2b) is in no way supported by DFT calculations, NMR identification or GC detection of intermediate stages or fragments. Thus, it is also not mechanically proven which bond is

broken exactly in the present model compounds. The experiments on transfer hydrogenation, which do not allow any conclusions to be drawn about the mechanistic contribution of the catalyst, are particularly unclear. In addition, the work already published on transfer hydrogenation with the Ru(triphos)tmm catalyst is not included in the analysis.

Additional comments:

- Comparison of the similarity of the lignin and epoxy resin structures is missing in the manuscript.
- There is no comparison to the previously published mechanism of C-C and C-O cleavage.
- The suggested bond cleavage mechanism must be verified by experiments (GC, NMR).
- Experiments with the addition of an acid co-catalyst are completely absent.
- For a publication with the vision of translation into application, experiments must be carried out on a larger scale.
- It must be shown what products are formed from the remaining polymer building blocks in addition to BPA.
- The structure of the dimeric ruthenium complex does not correspond to the prior art.
- The mechanism of the transfer hydrogenation and the role of (tmm) remains elusive.

In summary, the manuscript presents an interesting development in the field of C-O bond cleavage catalysis. However, using an existing catalytic concept previously demonstrated on lignin greatly reduces the novelty. Furthermore, the experimental data does not meet Nature's quality standards and the conclusions are not fully supported by the experimental data provided. In the reviewer's opinion, the present manuscript should be rejected. A publication in any journal can only take place if the state of the art is not obscured and the mechanistic proposals are substantiated by relevant experiments.

Referee #3 (Remarks to the Author):

The manuscript address a critical problem - the deconstruction of thermoset epoxy resins. As described in the introduction, many high performance composite materials (such as those in wind turbine blades) are made from thermosetting epoxide resins that currently have no practical end of life strategy and end up in land fills. The manuscript describes a novel ruthenium-catalyzed dehydrogenation-bond cleavage-reduction cascade to deconstruct the C-O bonds in fully cured epoxy. Chemical and visual evidence of the deconstruction of neat epoxy as well as an epoxy/fiber composite are provided in the manuscript and the extensive supporting information. While this is an exciting result and could be a promising strategy for recycling or upcycling of epoxy resins, the manuscript has several deficiencies and omissions. The manuscript is also difficult to read and written more for a chemistry journal than a broader/ multidisciplinary scientific audience.

1. No operando control experiment is reported for epoxy subject to the same time and temperature deconstructions conditions with no Ru catalyst present (NMR spectra after soak in toluene for 24 hrs at 160°C). Deconstruction is not expected but any (or lack of) chemical changes to the epoxy resin

under these conditions need to be reported.

2. Significant time and energy is needed for this deconstruction process, it is important to show the value of the recovered product and what functionality it has. What is the upcycling strategy? The manuscript does not describe or demonstrate how the recovered BPA and other products will be used.

3. A very large wt% of catalyst (ca. 6%) is used for the deconstruction process for a relatively small sample. Also, a significant amount of energy and time is needed to deconstruct a 100 mg sample. This leads to questions of the total cost and scalability - if the goal is to deconstruct an entire turbine blade. The manuscript needs to address the question of scale. How big a sample can be deconstructed with this strategy?

4. Varying yields of BPA were achieved. It seems problematic for a truly circular strategy that the waste product will contain BPA. What is the waste associated with the process (i.e. what is the highly viscous oil)?

5. The deconstruction of cured composites is exciting but the data in Figure 4 is insufficient to make conclusions about the fibers. What chemistry is left on the fiber surface? XPS analysis of recovered fibers seems warranted. Also, CT imaging does not have sufficient resolution to detect any damage to the fiber surface. Ultimately large enough sections of fibers need to be recovered and the tensile stress assessed.

Additional comments:

Figure 1a is confusing and hard to understand the the ball and stick schematic for the epoxy resin

FREC is not a commonly used acronym

Figures 3, 4c, S2, S3 (right), S4, is missing scale bars

Figure 4c is missing scale bars

Author Rebuttals to Initial Comments:

Referees' Comments and Our Responses

We would like to thank all three referees for their time, their comments, criticisms, and suggestions, which we feel has allowed us to modify the manuscript and provide new details to the work reported. Below we provide our answers to the questions and comments.

Referee #1:

1. The literature on selective cleavage of epoxy resins and their composites should be summarized and discussed in the introduction section of the manuscript. Otherwise, the science and technology advances of this work will be not clear for readers.

Answer: As suggested by the referee, the introduction was revised to highlight the importance and novelty of selectively disconnecting bonds in epoxy resins demonstrated here. Unfortunately, a comprehensive summary of reported works on the unselective cleavage of epoxy resins, such as pyrolysis or the use of modified resins containing breaking points would considerably lengthen the introduction. The introduction is therefore written in a way to give a conceptual overview providing context for the selectivity-based approach demonstrated in the manuscript, but of course including references to reviews and articles that elaborate on concrete methods for both reported approaches, as well as reuse perspectives (ref. 1–3, ref. 18 & 19, ref. 20 & 21, ref. 22–25). We hope that this will be satisfactory.

2. The chemical structure of benzophenone is much different from ketone 1. Whether more suitable model compound could be used to investigate the degradation mechanism? From the degradation mechanism proposed by authors, ketone should be produced, so it's better to directly capture the ketone during the degradation process.

Answer: It is a fair point that benzophenone is not similar to the corresponding ketone formed from the epoxy linkages. However, in that specific experiment, the focus lies on the dehydrogenation of isopropanol used as the hydrogen source under the catalytic conditions. Benzophenone was chosen in order to facilitate the study on the progression of the reaction by ^1H NMR spectroscopy. We agree with the referee that it would be of high interest to capture ketone intermediates, however, in general it can be challenging to capture true intermediates. Regarding this we would like to refer to the discussions on the kinetics requested below. While we could not detect the dehydrogenated epoxy **model 1** during these studies, we could detect the mono-cleaved ketone (**ketone III**) in trace amount with GC-MS, and showed in the following that synthesized **ketone III** can be disconnected in the catalytic protocol.

Please note that details and discussion of this experiment were moved to the Supplementary Information because to the points raised by referee #2.

3. For the real amine-cured epoxy networks, they should contain substantial amounts of tertiary amines and small amounts of secondary amines. Thus, besides the model **2** with secondary amines, a model with tertiary amines should be investigated. And for the anhydride-cured epoxy resins, beside the C-O bonds, there are a lot of ester bonds, do the ester bonds impact the disconnection efficiency of C-O bonds?

Answer: We thank the referee for pointing this out. As suggested, we prepared a new model compound (model **3** in the revised manuscript) with a tertiary amine and tested it in the catalysis.

Gratifyingly, this compound could also be deconstructed smoothly under the catalytic conditions. These results have been included into the revised manuscript and the Supplementary Information.

We were also able to acquire a sample of an anhydride-cured resin and to test it in the catalysis. Here, no BPA or other compounds could be recovered from the resin. Due to the different hardening pathway, the alcohol motifs of the linkages are functionalized as carboxylic esters. This could inhibit the dehydrogenation and thus eventually the C–O bond cleavage. These results and their discussion have been included in the revised manuscript on page 3.

4. The efficiency for the catalytic disconnection of C-O bonds is relatively low. Can it be improved ?

Answer: This is of course a good point. As stated in the manuscript, we consider this study primarily as a proof-of-concept discovery. While the presented method can be used to recycle end-of-life composite, as pointed out by the referee, the efficiency is low and as such commercial applications would be unlikely as it stands now. But a major advantage of homogenous catalysis is that a large range of modifications and optimizations on the catalyst can be undertaken and implemented. Many commercial processes, including those for hydroformylations or alkoxycarbonylations (which are all up to multi-million ton per year processes), have become increasingly efficient over decades of research on ligands, catalysts, and reaction conditions. Therefore, we do believe it is plausible that increased efforts on catalyst and ligand design could serve to enhance the efficiency of the presented protocol significantly. Nonetheless, as this view is speculative, we do not feel comfortable including such an outlook into the manuscript. Nevertheless, we do believe this discovery will open a new door for future research in epoxy and epoxy composite recycling.

Just to conclude, we have further examined the efficiency of the catalytic system on **model 1**, and we are happy to report that experiments with catalyst loadings as low as 0.38 mol% also led to quantitative formation of Me-BPA, corresponding to a TON of 533. These results are included in the revised manuscript on page 3.

5. For evaluation of the advance in disconnection of composites, beside the micromorphology and appearance of the reclaimed fibers, the chemical structures and the mechanical properties of the reclaimed fibers should be examined and compared with the virgin fibers. And it's better to compare with those from other recycle methods using sever conditions.

Answer: We thank the referee for pointing this out. As suggested, we conducted further analysis on fibers recovered from the wind turbine blade, as well as on the neat glass fibers as a reference sample. Gratifyingly, IR and XPS revealed that no organic residues remain on the fibers, but also that the sizing agents applied to the neat fibers have been removed. SEM supported these findings and revealed a smooth and clean fiber surface. Additionally, preliminary tensile testing was conducted, revealing comparable properties of recovered and neat fibers. These data have been added to the revised manuscript on page 7 and the Supplementary Information (Chapter 6.3). Unfortunately, for a full tensile testing study generating higher quality data, considerable efforts and better equipment beyond what is available to us would be required.

We agree that a comparison to fibers recovered via other methods such as pyrolysis would be highly interesting. Unfortunately, we do not have the set up to reproduce reported methods using such techniques.

6. The peaks in each NMR spectrum should be clearly matched with the chemical structures of the compounds.

Answer: As suggested, the lines in Fig. 2c and the corresponding spectra shown in the supplementary information (mechanistic sections) now correspond to the phosphines, protons or carbons in the complexes or compounds.

7. The stability of the catalysts during the reactions and their reuse should be investigated.

Answer: In accordance with this request, turnover number studies were conducted for the catalysis on **model 1** and Airstone. We were delighted to see that on **model 1** a TON of 533 was achievable over 16 h. For the more challenging thermoset powder, a TON of 105 over 4 days was achieved. These data have been included in the revised manuscript on pages 3 and 5.

Studies on catalyst reuse are unfortunately rather challenging. The catalyst in solution is sensitive to water/air, which makes separation of catalyst solutions from a product solution a complex problem. Admittedly, we have no good approach to tackling this question as of now. Therefore, we hope, that the data on the TON is satisfactory.

8. The disconnection yield after certain times were provided, but the kinetics of the disconnections were not systematically investigated, as a result, readers cannot see the balance point of the degradation reactions.

Answer: As suggested by the reviewer, we measured a kinetic profile on **model 1** over the course of 16 h with 2 h intervals. Gratifyingly, some interesting insights were obtained that support our mechanistic investigation:

From the revised manuscript (page 4): *“The kinetic profile of the bond disconnection on **model 1** was investigated (Fig. 2C), revealing an induction period of approximately 2 h, after which only a minute amount of Me-BPA could be detected via ¹H NMR spectroscopy and GC-MS analysis. However, after 4 h, a 50% yield of Me-BPA was observed with quantitative conversion being reached after 12 h. An induction period hints to catalyst activation preceding the catalytic cycle. At a reaction time of 4 h, traces of the mono-cleavage intermediate ketone **III** were detected via GC-MS. In all other entries only the starting material and product could be observed, indicating that the consumption of intermediates is significantly faster than that of **model 1**.”*

For full data, please check the revised Supplementary Information, section 3.3, page 23.

9. Authors used iPrOH and toluene as the solvent, I think different solvents might have very different degradation rate even degradation mechanism. Therefore, the effects of different solvents on the disconnection should be investigated.

Answer: Besides toluene, we tested 1,4-dioxane, 1,2-dimethoxyethane, dichloroethane and DMF as solvents on our “benchmark system” **model 1**. This information was included in Section 2.5, page 13 of the Supplementary Information of the original submission). However, as the C–O bond cleavage did not proceed efficiently in any of these solvent systems, we focused our studies on the use of toluene. Likewise, exchanging isopropanol for methanol or ethanol also led to very low reactivities on **model 1** (Section 2.3, page 12 of the Supplementary Information).

Referee #2:

1. In the present manuscript by Ahrens and Skrydstrup, a catalytic system for the depolymerisation of selected epoxy resins is described. The recycling of polymers is a field of current scientific interest and practical systems for the sustainable depolymerisation of complex materials remain rather elusive. The present approach tries to use molecular ruthenium catalyst systems for the effective catalytic disconnection of C–O Bonds in polymeric materials based on epoxy resins. The authors analysed the chemical structure of the epoxy resins and aimed at developing a transition metal catalysed method for disconnecting the C(alkyl)–O single bonds. The model compounds used by the authors are very similar to lignin model compounds, for which the cleavage of the C–O bonds has already been demonstrated. Especially in references 37 and 38 (incl. SI) the different bond cleavages and products are clearly broken down. The reference to these publications becomes even clearer when one recognizes that the catalyst Ru(triphos)tmm and its derivatives were also used in this preliminary work. Surprisingly, the authors avoid making a clear reference to this previous work and justify their own contribution with the low activity of the already published catalytic systems with the catalyst Ru(triphos)tmm.

Answer: We would like to apologize to the referee if we have given the impression that the previous work has not been properly referenced. This was definitely not our intention. Nevertheless, we are not entirely in agreement with the two criticisms brought forth by the referee, and we provide an explanation as well arguments. Two summarize briefly, the referee criticizes that

- a) the C–O bond cleavage reported here is the same as on lignin models, as the models are similar and should therefore behave the same.
- and b) there is no (clear) reference and comparison to the work on lignin given.

First of all, we would like to point out that the earlier reports on the lignin model studies applying Ru-catalyzed C–O bond cleavages are being referenced in the manuscript, and that a detailed comparison (experimental and discussion) has been made, which can be found in the Supplementary Information section of the first submission (pages 9 and 10). In the original manuscript, we write *“For the valorisation of lignin³⁵, homogenous ruthenium catalysis has been reported to disconnect C(alkyl)–O and C–C single bonds^{36–38}. Nonetheless, initial attempts to transfer the reported methodologies from lignin models to **model 1** (Fig. 1A), mimicking one of the most common motifs in epoxy resins, were unrewarding, leading either to low or no conversion (Table S1). However, after a comprehensive screening of potential catalysts and reaction conditions (Table S2–S6), an efficient protocol was identified.”*

In the original Supplementary Information section, we also presented the following subsection: *“2.1 Methodologies Reported for Lignin Valorization. For initial investigations of C–O bond cleavage on epoxy models, methodologies reported for the β-O-4 motif in lignin^{3,4} were considered (Fig. S5). The mechanism of the cleavage is assumed to go through acceptorless dehydrogenation of the alcohol moiety. Afterwards, the C–O bond adjunct to the ketone can be activated by the ruthenium catalyst. The liberated hydrogen is then consumed, liberating acetophenone, phenol and the active catalyst³.”*

Fig. S5. Ruthenium-catalysed C–O bond cleavage reported for lignin models.

Table S1. Initial attempts to transfer the reported procedures to epoxy **model 1** including some variations:

Entry	Conditions / Variations	Consumption	Me-BPA
Conditions A			
1	no variations	>0%	not detected
2	160 °C, 16 h	13%	traces
3	200 °C, 16 h, mesitylene as solvent	8%	15%
Conditions B			
4	no variations	>0%	not detected
5	160 °C, 16 h	8%	traces
6	160 °C, 16 h, triphos-Ru-TMM ^{a)}	23%	4%
Conditions C			
7	no variations	>0%	traces
8	160 °C, 16 h	13%	11%
9	160 °C, 16 h, bdepp-Ru-TMM ^{a)}	47%	27%

a) 5 mol% of isolated complex were used as catalyst instead of in situ forming the corresponding complex.

*It must be considered that the β -O-4 motif is based on a benzylic alcohol, while the epoxy motif is based on a secondary alkyl alcohol. Another major difference is that the epoxy **model 1** contains two aryl ether C–O bonds adjacent the alcohol, calling for the presence of an additional hydrogen source.”*

Firstly, we would like to stress that we clearly state in the manuscript and the Supplementary Information section, that we initiated our investigations based on the methods reported for lignin models. We never claimed that this is the first Ru-catalyzed C–O bond cleavage. We explain that we tested the reported procedures for lignin models and then optimized the catalysis conditions on an epoxy model compound until we had developed a suitable method as the quoted passages show. Even at increased reaction temperatures, the methods reported for the lignin models could not be transferred to a simple epoxy model. It is also worth mentioning that for epoxy compounds, the presence of isopropanol is absolutely crucial for the disconnection method presented here (see revised manuscript or explanation below), which is something unprecedented in the lignin-focused literature applying Ru-complexes. We discuss the chemical differences between the linkage motifs (benzylic vs alkyl alcohol, one vs two C–O bonds). While the argument is that these models are chemically the same, we clearly show that they do not behave the same at all in the catalysis. Furthermore, we provide a rationalization on this difference.

In order to control the length and focus of the manuscript, this and many other data / details were moved to the Supplementary Information section. We would gladly consider whether the detailed comparison should be part of the main text or the Supplementary Information. But based on the above discussion, we do not feel that it is fair to state that we are avoiding making a clear reference to this previous work.

Lastly, we would like to point out, that this criticism is focused on the very initial studies presented in the work here. **Model 1** is just one of the several linkages relevant for epoxy resins. The other motifs and especially the presence of amine functionalities diverge from what was studied on lignin models in literature. Not to mention that actual thermoset resins (as studied here) are molecularly highly complex and have different physical properties than simple model compounds. To the best of our knowledge, the methods from ref 37 and ref 38 were only tested on simple lignin models and never studied on model polymers or real lignin samples.

Finally, we include a comparison between **model 1** and lignin. Here, we have decided to expand the comparison in the revised Supplementary Information and include a scheme showing the structures side by side, based on the concern raised by the referee.

As taken from the revised Supplementary Information section on page 12: *“It can be concluded that despite superficial similarities, epoxy **model 1** and β -O-4 lignin models are not interchangeable in their chemical behaviour. It must be considered that the β -O-4 motif is based on a benzylic alcohol, while the epoxy motif is based on a secondary alkyl alcohol. The dehydrogenation of benzylic alcohols leads to conjugated ketones, which is not true for the epoxy motif. Another major difference is that the epoxy **model 1** contains two aryl ether C–O bonds adjacent the alcohol, calling for the presence of an additional hydrogen source.*

It is also noteworthy that on epoxy models, isopropanol deactivates bdepp-Ru-TMM, while it activates triphos-Ru-RMM (see Table S1 entries 6 and 8 vs. Table S2 entries 3 and 4)."

2. The main modification in the present manuscript is the addition of iPrOH, but the contribution of the secondary alcohol is not mechanistically proven by the present experiments, only simple dehydrogenation experiments are shown.

Answer: We thank the reviewer for pointing this out. In the original version of the manuscript, we had unfortunately omitted the discussion on the role of isopropanol. We have included our rationalization and the experimental data on the crucial role of isopropanol in the revised version of the manuscript:

Taken from the revised manuscript (pages 4 and 5): *"To study the catalyst activation during the induction period, operando monitoring experiments were conducted (Fig. 2E). In the presented method, isopropanol is used as an additional hydrogen source as certain epoxy linkages contain two C–O bonds adjacent to the central alcohol moiety. However, the deconstruction of **model 1** using 3 mol% triphos-Ru-TMM in toluene at 160 °C in absence of isopropanol yielded only 4% of Me-BPA after 16 h (see Table S1). If isopropanol only serves as an additional hydrogen source, approx. one half of the C–O bonds should have been cleaved. To scrutinise the role of isopropanol, the catalysis was set up in a J Young NMR tube, using 3 mol% of triphos-Ru-TMM and d^8 -toluene. ^1H and ^{31}P NMR spectroscopy were used to monitor the progress of the reaction. After 16 h at 160 °C, no conversion of the starting material could be observed. Likewise, triphos-Ru-TMM had not been consumed and no other ruthenium species could be detected. The reaction was repeated with 3 equiv of isopropanol present. Here, close to quantitative conversion of the starting material to Me-BPA was observed. At the same time, the starting ruthenium complex was consumed over the course of 16 h, and the formation of new signals corresponding to ruthenium species were detected both in the ^{31}P NMR spectra and the hydride region of the ^1H NMR spectra. A significant phosphine singlet peak at 47.3 ppm could not be assigned to a structure. A singlet peak at 43.4 ppm in the ^{31}P NMR spectra together with a broad singlet peak at –8.79 ppm in the corresponding ^1H NMR spectra could be identified as a hydride-bridged binuclear ruthenium(I) complex⁴⁸. This complex was then synthesized and tested as a potential catalyst on **model 1** under standard catalysis conditions. With no conversion being observed after 16 h, the formation of this species can be considered a deactivation pathway. Lastly, a triplet phosphine peak at 35.0 ppm with an integral of one and a doublet peak at 27.1 ppm with a corresponding integral of two, could be linked to a doublet of a doublet signal at –6.78 ppm in the ^1H NMR spectra using an ^1H , ^{31}P HSQC NMR experiment. Adding another equivalent of **model 1** to the reaction mixture containing the new ruthenium species and heating to 160 °C for another 16 h led to a clean deconstruction of the fresh substrate into methylated BPA. Interestingly, during the attempted deconstruction of **model 9**, the ruthenium species showing a hydride signal at –6.78 ppm formed selectively from the precatalyst. Single crystals suitable for X-Ray crystallography were obtained directly from the reaction mixture, allowing the identification of this species as triphos-Ru-H₂-CO⁴⁹. We propose that the CO originates from isopropanol, which presents a rare case of decarbonylation of a secondary alcohol⁵⁰. The carbonyl complex was then tested as the catalyst, but it proved inactive.*

These results strongly support the conclusion that triphos-Ru-TMM acts as a precatalyst, which is activated over an induction period with the unidentified species observed at 47.3 ppm in the ^{31}P NMR spectra being the resting states of the active catalyst. The presence of isopropanol is crucial for the activation of the precatalyst to take place, however, the mechanism of this process remains elusive."

3. In addition, the proposed mechanism (Fig. 2b) is in no way supported by DFT calculations, NMR identification or GC detection of intermediate stages or fragments. Thus, it is also not mechanically proven which bond is broken exactly in the present model compounds.

Answer: We agree with the comment provided by the referee. In order to gather further evidence for the proposed mechanism, we conducted kinetic studies (as proposed by referee 1), which allowed the detection of the proposed intermediate ketone **III** by GC-MS, albeit in trace amounts. We conducted an experiment using ketone **III** as the starting material, demonstrating that it too is consumed and forms Me-BPA. Furthermore, we conducted an experiment that allowed the detection and quantification of acetone as the corresponding disconnection product from **model 1** by ^1H NMR spectroscopy. We believe these results provide evidence for the mono-cleavage intermediate and the formation of acetone from the linkage motif supporting the C–O bond cleavage-based mechanism.

In the revised manuscript, we have included the following text on page 4: *“The kinetic profile of the bond disconnection on **model 1** was investigated (Fig. 2C), revealing an induction period of approximately 2 h, after which only a minute amount of Me-BPA could be detected via ^1H NMR spectroscopy and GC-MS analysis. However, after 4 h, a 50% yield of Me-BPA was observed with quantitative conversion being reached after 12 h. An induction period hints to catalyst activation preceding the catalytic cycle. At a reaction time of 4 h, traces of the mono-cleavage intermediate ketone **III** were detected via GC-MS. In all other entries only the starting material and product could be observed, indicating that the consumption of intermediates is significantly faster than that of **model 1**. DFT calculations revealed that the C–O bond dissociation energy of ketone **I** is 3.3 kcal/mol lower than that of ketone **III** (Fig. 2B), implying that the faster consumption of the latter is not controlled by C–O bond strengths. Possibly the second C–O bond activation is faster than the dissociation of the intermediate from the catalyst. Ketone **III** was synthesised and subjected to the catalytic conditions as the starting material in a one-to-one mixture with **model 1**. Both compounds were consumed yielding Me-BPA, which supports ketone **III** being an intermediate. To gather further support for the proposed C–O bond cleaving mechanism, efforts were undertaken to detect acetone as the corresponding disconnection product formed from the central linkage motif of **model 1** (Fig 2D). As the dehydrogenation of isopropanol also yields acetone, this hydrogen source was exchanged for 1 equiv of 1-phenylethanol. After 16 h of reaction time, the reaction vessel was cooled to rt and the crude mixture analysed using ^1H NMR and ^{13}C NMR spectroscopy. Both a quantitative formation of acetophenone from 1-phenylethanol and a 74% yield of acetone from the linkage motif were observed.”*

For the full data, please check the revised Supplementary Information, section 3.3 and section 3.4. While DFT calculations represent a powerful tool to investigate reaction mechanisms, we do not consider it to be a fully reliable method to analyze the catalytic cycle and of this transformation at this point in time. The active catalytic species or the resting state thereof have not been identified yet. Without knowledge on the chemical constitution of such species, we would believe the calculations could be considered speculative.

4. The experiments on transfer hydrogenation, which do not allow any conclusions to be drawn about the mechanistic contribution of the catalyst, are particularly unclear. In addition, the work already published on transfer hydrogenation with the Ru(triphos)tmm catalyst is not included in the analysis.

Answer: The major part on the experiments and their discussions on transfer hydrogenation have now been moved to the Supplementary Information. They serve to show that the dehydrogenation

of alcohols is viable under the applied reaction conditions. We considered the lack of consumption of triphos-Ru-TMM during the process as interesting enough to include in the manuscript. However, we see the point raised by the referee that it might not add much to the discussion on the deconstruction mechanism. We assume that the article on the triphos-Ru catalysed transfer hydrogenation of CO₂ by Klankermayer *et al.* is referred to here? We thank the referee for pointing out this oversight on our part. A reference to this article has been added to the revised manuscript (reference 47) and the revised Supplementary Information.

The following is stated in the revised manuscript (page 4): “*Interestingly, operando monitoring experiments using ¹H and ³¹P NMR spectroscopy, revealed that triphos-Ru-TMM is active for the dehydrogenation of isopropanol without forming other observable ruthenium species or metal-ligand cooperation (see mechanistic studies in the Supplementary Information section). This is noteworthy, as a study on transfer hydrogenation for the reduction of CO₂ using triphos-Ru-TMM, revealed a catalyst activation process in the presence of an acid co-catalyst⁴⁷.*”

47. Westhues, N. Klankermayer, J. *ChemCatChem* **11**, 3371–3375 (2019).

5. Additional comments:

- Comparison of the similarity of the lignin and epoxy resin structures is missing in the manuscript.

Answer: Please see our response above, the data is referenced in the manuscript and in detail in the Supplementary Information section. Additionally, the section in the supplementary information has been expanded (section 2.1)

- There is no comparison to the previously published mechanism of C-C and C-O cleavage.

Answer: We apologise for the oversight on our part: Regarding the C–O bond cleavage, a reference to relevant works has been added to the beginning of the discussion of the mechanism. The following is stated in the revised manuscript, page 4: “*As supported by the inertness of **model 5** to the catalytic conditions, and in line with mechanisms proposed for the cleavage on lignin β-O-4 linkage models^{36,38}, we postulate that the C–O aryl bond breakage is preceded by the dehydrogenation of the alcohol functionality in **model 1**, forming ketone **1** (Fig. 2A).*”

36. Nichols, J. M., Bishop, L. M., Bergman, R. G., Ellman, J. A. Catalytic C–O Bond Cleavage of 2-Aryloxy-1-arylethanol and Its Application to the Depolymerization of Lignin-Related Polymers. *J. Am. Chem. Soc.* **132**, 12554–12555 (2010).

38. vom Stein, T., Weigand, T., Merkens, C., Klankermayer, J., Leitner, W. Trimethylenemethane-Ruthenium(II)-Triphos Complexes as Highly Active Catalysts for Catalytic C–O Bond Cleavage Reactions of Lignin Model Compounds. *ChemCatChem* **5**, 439–441 (2013).”

We believe that it is important to point out, that the study reported in ref. 36 applies a Xantphos-Ru complex, resulting in a significantly different coordination sphere and catalytic species than on the triphos complexes used in our study. This makes a direct comparison in detail difficult and unrewarding.

Ref. 38 reports triphos-Ru-TMM to be active on lignin models for C–O bond cleavage but identifies bdepp-Ru-TMM to be more active. (Ref. 38 refers to bdepp as “triphos C” and triphos as “triphos

B"). In ref. 38, two mechanistic experiments were conducted using bdepp-Ru (not triphos) to show that dehydrogenation and transfer hydrogenation of alcohols/ketones on simple models are plausible. Experiments, data, calculations or proposals regarding catalyst activation or the catalytic cycle are neither included in the article nor in the corresponding Supporting Information section. To the best of our knowledge, no work following up with such studies has been published. Mechanistic studies on C–O bond cleavages (lignin or other models) using triphos-Ru complexes have thus not been reported, unlike implied in the review. Here, we believe it is also crucial to stress that on epoxy **model 1**, triphos and bdepp behave significantly different (see Table S1 and Table S2). While the triphos complex is activated by isopropanol, the corresponding bdepp complex is deactivated by isopropanol. This cycles back to the claim that the lignin models and epoxy models should be interchangeable, which they clearly are not despite superficial similarities.

Regarding the C–C bond cleavage as reported in ref. 37, while this cleavage type on lignin models is referenced to in the introduction of the original manuscript, it is not a mode of reactivity we have observed on epoxy models, resins or composites. Ref. 37 by Klankermayer, Leitner *et al.* provide detailed experimental studies on the C–C bond cleavage. These studies reveal that the presence of the methyl hydroxy group in lignin (marked in red in the scheme below) is necessary for allowing a mechanism cleaving the C–C bond (marked in blue). This also explains why ref. 36 and ref. 38 did not report C–C bond cleavage, as they use simpler model compounds omitting that specific methyl hydroxy group.

This functionality is not a part of any linkage motif found in epoxy, and therefore this reported reaction mode in ref. 37 is not a relevant mode for epoxy models, resins and composites. A reference to an alternative C–C bond activation mechanism as reported by Ozerov *et al.* on isopropanol (*Organometallics* **24**, 186–189 (2005)) has been added (ref. 50) and is discussed in the manuscript for the formation of an inactive carbonyl complex. However, this represents a catalyst deactivation pathway and therefore cannot be a productive part of the catalytic protocol. Here it is also important to note that the expected products of this pathway on **model 1** (namely dimethyl-BPA) were never detected by ¹H NMR spectroscopy or GC-MS analysis. It is more likely that this deactivation pathway consumes isopropanol. Based on this discussion, we consider it legitimate to omit a discussion on C–C bond cleavage as an alternative productive pathway for the work on epoxy deconstruction reported in this manuscript.

We would also like to point out that we have included mechanistic studies which support our postulated C–O bond cleavage mechanism in the revised manuscript in accordance with questions raised by the referees (please refer to the discussion provided above).

Finally, the bond dissociation energies of **model 1** and both **ketones I** and **III** calculated with DFT at the (U)M062X/6-311++(d,p) level of theory show that C–O bond cleavage is less energy demanding than for C–C bond cleavage. This data has been added to the Supplementary Information section 6.5.

- The suggested bond cleavage mechanism must be verified by experiments (GC, NMR).

Answer: Please see discussion above, in the revised manuscript we have included kinetic studies and an experiment confirming that acetone is being formed from the linkage motif, which both support the proposed C–O bond cleavage mechanism.

- Experiments with the addition of an acid co-catalyst are completely absent.

Answer: As suggested, four experiments were conducted using acid co-catalysts on **model 1**. Acid co-catalysts were found to deactivate the catalytic system. The entries have been included into the optimization part in the revised Supplementary Information section 2.6, page 15.

2.6 Screening of Structurally Related Catalysts and Acid Co-Catalysts

Table S6. Ruthenium complex screening.

Entry	Catalyst / Other Variations	Consumption	Me-BPA
1 (ref)	triphos-Ru-TMM	94%	85%
2	N-triphos-Ru-TMM	59%	43%
3	triphos-Ru-H ₂ -CO	4%	0%
4	triphos-Ru-HCl-CO	5%	traces
5	triphos-Ru-HCl-CO / 6 mol% KOtBu	29%	9%
6	triphos-Ru-TMM / 3 mol% HNTf ₂	13%	0%
7	triphos-Ru-TMM / 3 mol% MsOH	8%	0%
8	triphos-Ru-TMM / 3 mol% HNTf ₂ / H ₂ instead of i PrOH	4%	0%
9	triphos-Ru- H ₂ -CO / 3 mol% HNTf ₂	13%	traces

- For a publication with the vision of translation into application, experiments must be carried out on a larger scale.

Answer: As suggested, a scale-up experiment on a piece of wind turbine blade was conducted. A 5.13 g piece was used, which corresponds to a scale-up of 23 times with respect to the original entry. Gratifyingly, the deconstruction on this sample proceeded smoothly, and consistent yields were observed. This valuable information has been added to the revised manuscript (Fig 3B) and the Supplementary Information section.

- It must be shown what products are formed from the remaining polymer building blocks in addition to BPA.

Answer: As suggested the referee, a more elaborate discussion of the rest fraction has been included in the revised manuscript on page 5. Furthermore, in alignment with a point made by referee 3, a discussion on the potentials for reuse of this fraction has been added to the revised manuscript (page 7). In addition to ¹H NMR and MALDI-TOF MS data, an IR spectrum was included in the revised version. Unfortunately, discreet chemical structures could not be determined due to the complexity of the mixture.

From the revised version of the manuscript: “Furthermore, a highly polar rest fraction could be recovered. Analysis of this fraction by MALDI-TOF MS, ¹H NMR and IR spectroscopy disclosed a complex mixture of oligomers, containing alkyl chains, alkyl ethers, amines and small amounts of aromatics. [...] The rest fraction consisting of different oligomers can unfortunately not be used as chemical building blocks. Nonetheless, valorisation strategies beyond energy recovery can be envisioned. For example, pyrolysis has been demonstrated to process mixed plastic wastes, including nitrogen containing polymers, into e.g. naphtha equivalents or syngas^{51,52}. As such, this rest fraction may find uses as a carbon feedstock source for the chemical industry.”

- The structure of the dimeric ruthenium complex does not correspond to the prior art.

Answer: As suggested by the referee, the Ru–Ru bond has been removed in the revised manuscript and the Supplementary Information.

However, we would like to point out that while some articles do not show a Ru–Ru bond in this specific complex (Klankermayer, Leitner *et al. J. Am. Chem. Soc.* **2014**, *136*, 38, 13217–13225; Beller *et al. J. Am. Chem. Soc.* **2015**, *137*, 42, 13580–13587.), other articles do show the structure with a Ru–Ru bond (Klankermayer, Leitner *et al. Chem. Sci.* **2015**, *6*, 693–704; Cole-Hamilton *et al. Chem. Eur. J.* **2013**, *19*, 11039–11050.)

- The mechanism of the transfer hydrogenation and the role of (tmm) remains elusive.

Answer: While it is true that we cannot offer a sound mechanistic proposal for the dehydrogenation or transfer hydrogenation of alcohols by triphos-Ru-TMM at this point, we do report the mechanistic insights we obtained from operando monitoring experiments (in the revised version moved to the Supplementary information section 3.1 and 3.2), identifying that triphos-Ru-TMM itself is the resting state of the transfer hydrogenation catalyst.

Furthermore, as we show in the operando monitoring experiments on the epoxy model, triphos-Ru-TMM itself is a precatalyst for the C–O bond cleavage studied. In the catalytic cycle (including dehydrogenation) with the epoxy model, the active species is another ruthenium complex of unidentified structure as reported in the manuscript. Therefore, we believe that a full understanding of the triphos-Ru-TMM catalyzed dehydrogenation or transfer hydrogenation of alcohols goes beyond the scope of this work and will require a more detailed study, way beyond the focus on the chemistry in epoxy models, resins and composites presented here. Nevertheless, we agree with the referee that such important studies will be necessary for increasing the efficiency of the catalytic system for epoxy deconstruction.

Referee #3:

1. No operando control experiment is reported for epoxy subject to the same time and temperature deconstructions conditions with no Ru catalyst present (NMR spectra after soak in toluene for 24 hrs at 160°C). Deconstruction is not expected but any (or lack of) chemical changes to the epoxy resin under these conditions need to be reported.

Answer: As suggested by the referee, we heated the powdered Airstone 760E/766H in toluene with isopropanol for 24 h. Afterwards, we analyzed the solution but could not detect any compounds

liberated from the polymer. This information has been added to the revised manuscript. Unfortunately, we do not have the analytical equipment to analyze the left over Airstone particles regarding chemical changes in the solid polymer. Therefore, we focused on the solvent phase. The following text has been modified in the revised manuscript on page 5. *“Initially, a “dogbone” of the cured resin was grounded into powder, suspended in toluene containing 8 vol% isopropanol and stirred at 160 °C in the absence of a catalyst (entry 1). After 24 h, the reaction was stopped, and the solution analysed by ¹H NMR spectroscopy and GC-MS. No compounds liberated from the resin could be detected.”*

The corresponding control experiment was also conducted on a piece of composite from the wind turbine blade. Here too, no organic compounds could be detected after 3 d of heating in toluene and isopropanol. Furthermore, no fibers were liberated according to visual inspection. This was included in the manuscript and the visual evidence added to the Supplementary Information. The following text in the revised manuscript on page 6 has been added: *“To confirm the role of the catalyst, a control experiment was conducted. A piece of the turbine blade was heated in toluene with isopropanol for 3 d at 160 °C in the absence of a ruthenium complex. No organic compounds could be detected by ¹H NMR spectroscopy and GC-MS in the toluene phase. Furthermore, neither the liberation of fibers nor of the metal grid from the composite could be observed (see Supplementary Information section).”*

2. Significant time and energy is needed for this deconstruction process, it is important to show the value of the recovered product and what functionality it has. What is the upcycling strategy? The manuscript does not describe or demonstrate how the recovered BPA and other products will be used.

Answer: As suggested by the referee, a discussion on the reuse potential for recovered BPA, rest fraction and fibers with corresponding references has been added to the manuscript. The following text can be found on page 7 of the revised manuscript: *“For the components recovered from the end-of-use composites, perspectives for circularity can be considered. The high purity of the BPA recovered in principle allows its reuse in established production chains for epoxy resins, polycarbonates or polyesters, replacing virgin BPA produced from naphtha feedstock. The rest fraction consisting of different oligomers can unfortunately not be used as chemical building blocks. Nonetheless, valorisation strategies beyond energy recovery can be envisioned. For example, pyrolysis has been demonstrated to process mixed plastic wastes, including nitrogen containing polymers, into e.g. naphtha equivalents or syngas^{51,52}. As such, this rest fraction may find uses as a carbon feedstock source for the chemical industry. Lastly, for the glass and carbon fibers, which are recovered in high quality, several reuse approaches have been reported. This includes the use of recovered fibers to construct new composite materials after a chemical sizing or re-sizing process^{53,54}. With these considerations in mind, our catalytic process can be considered as a proof-of-concept demonstration that a circular economy may well be achievable for these valuable and relevant materials.”*

51. Eze, W. U., Umunakwe, R., Obasi, H. C., Ugbaja, M. I., Uche, C. C., Madufor, I. C. Plastics waste management: A review of pyrolysis technology. *Clean Technol. Recycl.* **1**, 50-69 (2021).

52. Kijo-Kleczkowska, A., Gnatowski, A. Recycling of Plastic Waste, with Particular Emphasis on Thermal Methods - Review. *Energies* **15**, 2114–2135 (2022).

53. Rani, M., Choudhary, P., Krishnan, V., Zafar, S. A review on recycling and reuse methods for carbon fiber/glass fiber composites waste from wind turbine blades. *Compos. Part B: Eng.* **215**, 108768–108783 (2021).

54. Gonçalves, R. M., Martinho, A., Oliveira, J. P. Recycling of Reinforced Glass Fibers Waste: Current Status. *Materials*, **15**, 1596–1614 (2022).

3. A very large wt% of catalyst (ca. 6%) is used for the deconstruction process for a relatively small sample. Also, a significant amount of energy and time is needed to deconstruct a 100 mg sample. This leads to questions of the total cost and scalability - if the goal is to deconstruct an entire turbine blade. The manuscript needs to address the question of scale. How big a sample can be deconstructed with this strategy?

Answer: As suggested by this referee and also referee 2 (see point 5), a scale-up experiment on a piece of wind turbine blade was conducted. A 5.13 g piece was used, which corresponds to a scale-up of 23 times with respect to the original entry. Gratifyingly, the deconstruction on this sample proceeded smoothly, and consistent yields were observed. This valuable information has been added to the revised manuscript (Fig 3B) and the Supplementary Information.

Unfortunately, moving on to larger scales will not be possible for us as we do not have the appropriate equipment that allows us processing larger pieces of the composite. It is a fair point that the process is costly as it is now with a relatively high catalyst loading. However, the catalysis on several gram scale does in principle demonstrate the deconstruction approach is scalable to significantly larger pieces. Regarding the economic feasibility, we believe that a full LCA / industrial implementation strategy on this process would go beyond the scope of this initial proof-of-principle work. A substantial effort afterwards will be necessary to optimize the approach to more industrially viable scales.

4. Varying yields of BPA were achieved. It seems problematic for a truly circular strategy that the waste product will contain BPA. What is the waste associated with the process (i.e. what is the highly viscous oil)?

Answer: It is true that the rest fraction still contains BPA motifs. However, it is important to point out that this fraction is not contaminated with free BPA (detection level of GC-MS analysis). The BPA building blocks are covalently bound within the oligomers and do not possess the same properties and mobility as free BPA. Furthermore, epoxy linkages are not known to leak BPA to the best of our knowledge. As suggested (also in alignment by a point raised by referee 2), a more elaborate discussion of the rest fraction has been included in the revised manuscript. In addition to ¹H NMR and MALDI-TOF MS data, an IR spectrum was included in the revised version.

From the revised version of the manuscript on page 7: *“Furthermore, a highly polar rest fraction could be recovered. Analysis of this fraction by MALDI-TOF MS, ¹H NMR and IR spectroscopy disclosed a complex mixture of oligomers, containing alkyl chains, alkyl ethers, amines and small amounts of aromatics.”*

5. The deconstruction of cured composites is exciting but the data in Figure 4 is insufficient to make conclusions about the fibers. What chemistry is left on the fiber surface? XPS analysis of recovered fibers seems warranted. Also, CT imaging does not have sufficient resolution to detect any damage

to the fiber surface. Ultimately large enough sections of fibers need to be recovered and the tensile stress assessed.

Answer: We thank the referee for pointing this out together with referee #1. As suggested, we conducted further analysis on fibers recovered from the wind turbine blade, as well as on neat glass fibers as a reference sample. Gratifyingly, IR and XPS revealed that no organic residues remain on the fibers, but also that the sizing agents applied to the neat fibers have been removed. SEM supported these findings and revealed a smooth and clean fiber surface. Additionally, preliminary tensile testing was conducted, revealing at comparable properties of recovered and neat fibers. These data have been added to the revised manuscript on page 7 and the Supplementary Information. Unfortunately, for a full tensile testing study generating higher quality data, considerable efforts and better equipment beyond what is available to us would be required.

6. Additional comments:

- Figure 1a is confusing and hard to understand the ball and stick schematic for the epoxy resin

Answer: For clarification we added a specification for this scheme in Fig 1. The following text has been modified for Fig. 1 in the revised manuscript: *“a, Schematic illustration of a crosslinked epoxy resin matrix and molecular structures of linkage motifs, the blue circles represent linkage sections while black lines represent polymer branches. The carbon–oxygen bonds adjacent to BPA (red), are targeted in order to deconstruct the polymer matrix.”*

- FREC is not a commonly used acronym

Answer: We decided to use FREC as an abbreviation in order to specify that we are specifically examining Epoxy-based Fiber-reinforced polymer composites. Unfortunately, we could not find a common acronym for epoxy composites in the literature. In the revised manuscript and the Supplementary Information section, we have replaced FREC by “epoxy composite” or “composite”, as suggested.

- Figures 3, 4c, S2, S3 (right), S4, is missing scale bars

Answer: For Figure 3 and S2–4, unfortunately no photos with a scale bar were taken before disassembling the pieces. In the revised manuscript, the sizes of the disassembled pieces being between 1 cm to 1.5 cm in length and their width have been added including to the Supporting Information section (description of Fig.3, S2–4). Additionally, the weight of each piece has been added to Fig.3 and S2–4.

- Figure 4c is missing scale bars

Answer: Please note that there was a mistake in the original submission of Fig 4 (two Fig. 4c). The relevant part is referred to as Fig. 4b in the revised manuscript.

The 3D renderings of Fig. 4b are shown in perspective, which means that the scale depends on the positioning in the image. Therefore, we decided to omit the scale bars and rather refer to the scale bars in the 2D views. A remark has been added to the figure description in the revised manuscript:

“b, 3D renderings of reconstructed image stacks showing fiber organisation in 3D; gray levels corresponding to air have been made transparent; for scale refer to the 2D slices in a.”

Reviewer Reports on the First Revision:

Referee #1 (Remarks to the Author):

Authors have revised the manuscript according to the improvement suggestions. Although some of the issues pointed out by reviewers were not addressed, authors gave rational response. As described by authors, this work is a proof-of-concept discovery, one work cannot achieve perfect results and real application. With this discovery, I think more researchers on catalysts or materials recycling would further improve the efficiency, economy and ecology in the future. After comprehensively consideration, I recommend this work to be published in Nature.

Referee #4 (Remarks to the Author):

Authors have adequately addressed reviewer comments